# Semantic Self-Images and Well-Being in Young and Older Adults: Does the Accessibility Matter?

**DOI:** 10.3390/brainsci12060716

**Published:** 2022-05-31

**Authors:** Manila Vannucci, Carlo Chiorri, Claudia Pelagatti, Laura Favilli

**Affiliations:** 1Department of NEUROFARBA, Section of Psychology, University of Florence, Via San Salvi 12, Padiglione 26, 50135 Florence, Italy; laura.favilli1@stud.unifi.it; 2Department of Educational Sciences—Psychology Unit, University of Genoa, Corso Podestà 2, 16128 Genova, Italy; 3Department of Education, Languages, Intercultures, Literatures and Psychology, University of Florence, 50121 Florence, Italy; claudia.pelagatti@unifi.it

**Keywords:** autobiographical memory, aging, self, identity, well-being

## Abstract

In the present study we investigated whether and how age group, dimensions of well-being and their interactions predicted the phenomenological properties of semantic self-images, taking also into account the different levels of accessibility of self-images (i.e., order of generation). Results on the first self-image revealed that, independently of age, higher levels of life satisfaction predicted higher likelihood of positive than negative statement and higher levels of negative affect and life satisfaction predicted higher levels of personal relevance of the self-image. When all self-images were considered, for higher levels of life satisfaction neutral and positive self-images were more likely than negative ones, and for lower levels of positive affect, neutral images were more likely than negative ones. Moreover, young adults were more likely than older adults to report neutral rather than negative self-images and, for higher levels of positive affect, they were more likely to report neutral and positive images instead of negative ones. These results suggest that the accessibility of semantic self-images should be taken into account in the investigation of the complex association between well-being and semantic self-images. Theoretical and practical implications of these results are discussed.

## 1. Introduction

One of the most influential models in the field of autobiographical memory is the Self-Memory System (SMS) [1,2], a conceptual framework that has clarified the interconnectedness and the bidirectional influence between the self and autobiographical memory. Although in psychology the term “self” has been used to refer to a wide variety of constructs and definitions, including both implicit (e.g., bodily perception) and explicit aspects (e.g., personality traits, self-beliefs) [3,4,5], in the SMS model, a key role is played by the declarative kinds of self-knowledge (conceptual self) [6], that is, knowledge that people have about themselves, also referred to as “semantic self-images” [7].

In this regard, a number of studies on adult normal populations have consistently shown that semantic self-images, which reflect semantic representations in autobiographical memory [8], have an organizing role in autobiographical retrieval [7,9,10] and in constructing future autobiographical events [9,11,12].

Moreover, as we briefly review below, several studies have shown that the investigation of the phenomenological properties of semantic self-images has revealed changes in self-knowledge associated with aging, as well as identity alterations in individuals with brain damage, neuropsychiatric, and psychopathological disorders.

Interestingly, recent evidence [13] has also suggested that the emotional valence of semantic self-images (i.e., how people view themselves at the semantic level) was more closely related to well-being than the valence of episodic autobiographical memories (i.e., how people view themselves at an event-specific level), particularly in older adults.

In the present study, we capitalize on these findings and aim to further investigate how age group (i.e., young vs. older adults), dimensions of psychological well-being, and their interactions can predict the phenomenological properties (i.e., emotional valence, personal relevance, category content) of semantic self-images, taking also into account the different levels of accessibility of self-images (i.e., order of generation of self-images/self-statements). Somewhat surprisingly, this issue has largely been overlooked in the field. However, as we review in the last part of Section 1, the results of the very few studies on the accessibility of semantic self-images [14,15,16] are quite promising and have paved the way for further investigations.

### 1.1. Phenomenology of Semantic Self-Images

Semantic self-images, which are typically assessed by employing self-image generation tasks (e.g., production of statements starting with the words “I am”), have been frequently used as a measure of identity because they are thought to capture different aspects of the self-knowledge component of identity. These aspects can be investigated along different phenomenological dimensions, such as content categories, emotional valence, and personal relevance. 

As for content categories, self-images may include self-knowledge of one’s traits (e.g., “I am outgoing”, “I am shy”), or they may refer to more concrete dimensions, such as physical appearance (e.g., “I am blonde”), or social constructs, such as roles or group membership (e.g., “I am a student”; “I am a mother”). This heterogeneity is supposed to reflect the multi-faceted nature of the self [17].

Despite a variety of coding systems, with different numbers of content categories, being developed and used in the studies to classify the statements/self-images (e.g., consensual vs. sub-consensual [17]; traits, social identities, specific attributes, evaluative descriptions, physical descriptions, emotional states, peripheral information, global descriptions [18]; concrete vs. abstract [7]; psychological, social, and physical [19,20]), the investigation of the responses has consistently revealed differences in self-images across cultures [18,21], gender [22], and age [23].

As for aging, the study by McCrae and Costa [23] revealed that young people were more likely to report personality traits, family roles, and personal relationships, whereas older people were more likely to report age, health, life circumstances, interests, hobbies, and beliefs. 

Alterations of self-knowledge have been consistently reported in neuropsychological patients with amnesia [24,25], Alzheimer’s disease [26], Parkinson’s Disease [27], Asperger’s syndrome [28] and psychiatric and psychopathological disorders such as schizophrenia [29], PTSD [30] and social anxiety disorder [31] (for a review, see Ernst and Rathbone [32]).

Importantly, Rathbone and Moulin [33] have recently developed and published a database of self-image norms, based on the coding of 2412 self-images, to provide a clear indication of the most frequently cited self-images for people in a specific age-range and for a specific gender, which is of great utility for comparisons with clinical and neuropsychological samples. 

As for the investigation of the emotional valence of semantic self-images, studies on adult normal populations revealed that people tend to describe and view themselves in a positive light [10,13,33,34] and a predominance of positive self-images has been reported in both young and older adults [13,35]. 

In this regard, a recent study by Rathbone et al. [13] has shown that the emotional valence of semantic self-images was linked to measures of well-being with a stronger pattern of association than for the valence of episodic autobiographical memories. Moreover, this association was more pronounced in older than younger adults: specifically, in the group of young adults, more positive semantic self-images were significantly associated with higher levels of positive affect, global psychological well-being, and life satisfaction (with *r* ranging from 0.38 to 0.46), while in older adults the valence of semantic self-images significantly correlated with all measures of well-being, with more positive self-images being associated with increased levels of positive affect, well-being, life satisfaction, optimism, and lower levels of negative affect (with *r*, ranging from 0.49 to 0.75). Showing that well-being is closely associated with how positively participants perceive and define themselves, and that this is especially true for older adults, has important applied implications for promoting well-being in aging [13].

In this pioneering study, the semantic self-images produced by participants were considered all together, without taking into account the different levels of accessibility of the images, namely the order of generation of the different statements. That leaves open the question of whether and how aging and well-being might affect in a different way the phenomenological properties of semantic self-images with a different level of accessibility. Moreover, in the study by Rathbone et al. [13] the interaction effects between age group and measures of well-being were not formally nor directly tested.

### 1.2. Accessibility of Semantic Self-Images

The issue of the accessibility of semantic self-images is of theoretical importance. Although the self-concepts are considered stable, different aspects of the self can be differently salient at different times and/or in different contexts [36,37,38,39], so that different types of descriptors (semantic self-images) are generated in self-statements tasks. For example, Isbell et al. [36] found that the extent to which individuals describe themselves at the “Twenty Statements” Test (TST) [17] in global/abstract terms (e.g., “I am honest”) versus more specific/concrete ones (“I am hungry”) depends on how one feels at the moment (i.e., happy, sad, angry, fearful): participants in predominantly happy and angry states used more abstract statements to describe themselves than did participants in predominantly sad and fearful states. Although all the self-generated statements at TST are supposed to be descriptive of the working self, these statements differed in their levels of accessibility, with the attributes generated earlier in the sequence being more accessible compared to those generated later in the series. 

If so, one might argue that the order of generation, which reflects the different level of accessibility, should be taken into account when the phenomenological properties of self-images are considered and when comparisons between groups (e.g., young adults vs. older adults) are carried out. In their seminal paper on TST, Kuhn and McPartland [17] indicate that “*the ordering of responses is a reflection of the make-up of the self-conception*” (p. 72) and that “*in the ordering of responses we are dealing essentially with the dimension of salience of self attitudes*” (p. 72). From the order of the statements, the authors could show that participants tended to report first concrete or consensual self-concepts, which refer to roles or group membership (e.g., “I am a student”; “I am a mother”), and only later (if at all) any abstract or sub-consensual statements, reflecting beliefs or traits (e.g., “I am shy”, “I am interesting”). 

Somehow surprisingly, the issue of the different level of accessibility of semantic self-images has largely been overlooked in the field, and, as far as we know, only a very few studies [14,15,16] have successfully addressed it. For example, in a cross-cultural study, Carpenter [15] examined whether statements generated earlier in the sequence at TST task were rated as more descriptive by participants. To this aim, the 15 statements were divided into first series (attributes 1–5), second series (attributes 6–10) and third series (attributes 11–15). The comparisons revealed that statements generated earlier in the sequence (series 1 and 2) were rated as significantly more descriptive than those in the later series (series 3). 

In a more recent study, Rathbone and Moulin [16] tested the hypothesis that more readily accessible self-images were evaluated as more important, and they were more effective in cueing autobiographical memories. In the study, participants were first asked to perform the TST and then the 1st, 5th, 10th, 15th, and 20th self-generated “I am” statements were each used as cues for a two-minute autobiographical memory task. Following the task, participants were asked to rate each of the 20 statements for personal significance (i.e., how important and central each statement was in defining their sense of identity). The authors found that identity statements in the first position, that is, self-images that first came to mind, were able to trigger significantly more memories than the others. Moreover, a negative association (*r* = −0.23) between the serial position of a statement (from 1 to 20) and its the personal significance was found: earlier generated statements were evaluated as more important aspects of the self.

## 2. The Present Study 

In the present study, we capitalize on the few but promising findings on the accessibility of semantic self-images and test whether and how age group (i.e., young vs. older adults), dimensions of psychological well-being (i.e., negative and positive affect, perceived stress, and satisfaction with life) and their interactions predicted the phenomenological properties (i.e., emotional valence, personal relevance, category content) of semantic self-images, taking into account different levels of accessibility of semantic self-images (i.e., the ease with which different self-images come to mind).

Specifically, we examined whether self-concepts that come to mind first (i.e., the first self-statements), which are the most salient and accessible self-images, and the ones which are not conditioned on previous answers differ from the other images in terms of phenomenological properties and patterns of influence by aging and dimensions of well-being. 

To this aim, we specified prediction models for the first self-image, for all self-images regardless of their order of generation, and for all-self-images taking into account their order of generation. Since the data were collected in a period of moderate COVID-19 related restrictions, we asked participants to complete also a measure of Fear of COVID-19 and we asked them whether they had been infected with it. 

## 3. Materials and Methods

### 3.1. Participants

Participants were 47 young adult undergraduates of psychology at the University of Florence and 37 older adults enrolled in Psychology courses at the “Università dell’Età Libera” of Florence, all native Italian speakers. See Table 1 for details about background variables. They were invited to take part in the study through an advertisement sent to the mailing list of the course they were attending that indicated the link to access the survey described in the Section 3.3. Globally, 101 and 92 individuals connected to the link to access the survey for the first and second session. However, 11 participants did not complete the first session, and two participants did not complete the second session. Of the remaining 89 participants, we excluded those who reported to be between 30 and 60 years old and those older than 75, and a participant who did not provide enough answers, thus leaving the final 84 cases whose data were actually used for the analyses (Table 1).

All participants volunteered to participate after being presented with a detailed description of the procedure and they were treated in accordance with the Ethical Principles of Psychologists and Code of Conduct [40]. The study was conducted according to the guidelines of the Declaration of Helsinki [41] and approved by the Ethics Committee of the University of Florence (protocol code n. 0107292, date of approval 28 July 2020). To be included in the study, participants had to report to be at least 18 years old and they did not receive any compensation for their participation.

### 3.2. Materials

Since data collection was run in a period of moderate COVID-19 related restriction in Tuscany (where the research was carried out) we took into account the possible role of COVID-related variables. Specifically, we asked participants whether they had been infected with it (no one reported being infected) and to complete the measure of Fear of COVID-19 described below. Descriptive statistics and results of the comparisons of scale scores between age groups are reported in Table 1.

Assessment of well-being. Participants completed the following questionnaires: -*Fear of COVID-19 Scale* (FC-19) [42,43]. The FC-19 is a 7-item scale that assesses COVID-19-related fear, worry, and anxiety. Items are rated on a 5-point, Likert-type, agreement scale ranging from “strongly disagree” to “strongly agree”. Higher scores indicate higher levels of fear of COVID-19.-*Satisfaction with life scale* (SWLS) [44,45]. The SWLS is a 5-item measure of global life satisfaction, intended as the cognitive component of subjective well-being. Items are rated on 7-point, Likert-type, agreement scale “strongly disagree” to “strongly agree”. Higher scores indicate higher levels of satisfaction with life.-*Positive and Negative Affect Schedule* (PANAS) [46,47]. The PANAS consists of two 10-item, self-report adjective checklists, one assessing positive affect (i.e., excited, interested, PANAS-P) and the other one assessing negative affect (i.e., upset, scared, PANAS-N). Participants were asked to rate how much they have experienced each emotional state in the last month on a 5-point, Likert-type, intensity scale “very slightly” to “extremely”. Higher scores indicate higher levels of positive and negative affect.-*Perceived Stress Scale-short version* (PSS-10) [48,49]. The PSS-10 is a 10-item, self-report measure of perceived stress levels. Participants are asked to report on the frequency of thoughts and feelings related to stressful events that they experienced in the previous month. Each item is scored on a 5-point, Likert-type, frequency scale ranging from “never” to “very often”. Higher scores indicate higher levels of perceived stress.

Assessment of semantic self-images. To assess semantic self-images, we asked participants to complete a short version of the “Twenty Statements” Test (TST) [17]. The TST is a task in which participants are asked to generate a set of “I am” statements describing themselves. The TST has been successfully used in several different fields, i.e., to investigate the alteration of self-knowledge in brain-damaged patients [24,25,26], and psychiatric and psychopathological disorders [29,30,31].

Although the original version of TST includes 20 statements, shorter versions with as few as 7 or 10 responses have been proposed, and some studies suggested that they are even more effective [50,51]. Moreover, some authors noticed that many participants tended to give up after 10 items and, if forced to go on, they might either repeat previous sentences or produce trivial responses. Therefore, we limited to ten the number of statements to report.

We told participants to complete each statement as soon as it popped in their mind (if nothing came to their mind, they could skip forward). After writing their answer, they also had to specify the emotional valence (7-point rating scale; from “very negative” to “very positive”) and the personal relevance (5-point rating scale; from “It reveals nothing” to “It reveals very much”) of the statement.

### 3.3. Procedure

Participants were tested online, in two sessions. They received an email with a link to a web page, for each session. We used the Limesurvey platform for designing and administering the survey. In both sessions, participants were asked to complete a form that comprised background information (about gender, age, and education) and they were also prompted to generate their personal ID, answering some questions (not reported here to ensure confidentiality). In the first session we assessed the dimensions of well-being, while in the second one the phenomenological properties of semantic self-images. Also included among our measures were other questionnaires that were of interest to other researchers and research questions, and they were not analyzed for the present work. To avoid priming particular aspects of the self, the two sessions were administered two days apart. The task was self-paced. The completion of the survey required minimal computer/smartphone proficiency and all participants reported being familiar with computers, smartphones, emailing, and online learning platforms. 

### 3.4. Coding of the Statements

Before performing the analyses, all statements reported by participants were coded by two independent judges as belonging to different statements categories. We based our classification on the categories used in previous studies [18,52]. Specifically, responses were coded into eight theme categories, namely traits (e.g., “I am shy”, “I am determined”), social identities (e.g., “I am a grandmother”, “I am a student”), specific attributes (e.g., “I am a book lover”, “I am a dog lover”), evaluative descriptions (e.g., “I am good at cooking”, “I am lucky for being surrounded by people who love me”), physical descriptions (e.g., “I am diabetic”, “I am fat”), emotional states (e.g., “I am sad”, “I am happy”), peripheral information (e.g., “I am tired”, “I am far away from my family”), and global descriptions (e.g., “I am an individual”, “I am myself”). In line with previous studies, we also identified a separate category for uncodable responses (i.e., repeated or nonsense statements, e.g., “I am a river”, “I am a tree”) and excluded these statements from the analyses on thematic contents. After completing the categorisation, we computed the inter-rater agreement between the coders and it resulted to be almost perfect (Cohen’s kappa = 0.82, *SE* = 0.02). Minor disagreements were solved by discussion.

## 4. Results

Young and older adults did not significantly differ in the levels of Fear of COVID-19 (*p* = 0.534, *d* = −0.14). Given that we had found that the age groups were unbalanced for negative affect (PANAS-N) and perceived stress (PSS) scores (Table 1), to get correct estimates of the age group differences on the other variables we used propensity score analysis [53] to obtain weights to be applied to observations in subsequent analyses. After matching, the standardized differences (in Cohen’s *d* metric) were <0.01 and 0.12 for PANAS-N and PSS, respectively, indicating an acceptable balance (i.e., *d* < 0.20). Note that, due to missing values in the covariates, one participant from the young group and seven participants from the elderly group had to be excluded from the analyses.

In order to test whether age group, well-being dimensions and their interactions predicted the emotional valence, the personal relevance, and the category content of self-images, intercept-only multilevel regression models were specified. These analyses were carried out: (i) on the first self-image alone; (ii) on all self-images, independently of the order of generation but taking into account the nesting of observations in participants and statements; (iii) on all self-images, taking into account the order of generation by specifying the fixed effect of statement, i.e., a 10-category factor. 

Globally, older participants reported more statements (*M* = 9.53, *SD* = 1.44, range = 3–10) than young participants did (*M* = 8.64, *SD* = 1.89, range = 4–10; *t*(81.93) = 2.44, *p* = 0.017, *d* = 0.53 [0.09, 0.96]).

### 4.1. Emotional Valence

Previous studies considered the emotional valence rating as a metric variable in which higher scores corresponded to more positive emotional valence. However, the ends of a bipolar scale such as the one used in this study express the same intensity of an opposite valence (i.e., negative vs. positive) while central values indicate a neutral (i.e., neither positive nor negative) valence. Therefore, it does not seem fully appropriate to consider the valence scores as different levels of intensity of a single dimension. Recent studies have pointed out the issues of using scores from bipolar scales as unidimensional, as most likely they are not [54,55,56]. We thus recoded the valence scores in three categories: Negative (“very negative” or “fairly negative”), Neutral (“a bit negative, neither positive nor negative; “a bit positive”), and Positive (“fairly positive” and “very positive”). The valence of the self-images on each statement of the TST is reported in Table 2. Globally, positive and neutral self-images had 3.19 (*p* < 0.001) and 2.11 (*p* < 0.001) times the odds of negative self-images of being reported, and positive self-images had 1.31 (*p* = 0.086) times the odds of neutral self-images of being reported, thereby showing that positive self-images tended to be predominantly reported.

When we considered the first self-image, positive self-images had 4.10 (*p* < 0.001) and 2.73 (*p* < 0.001) times the odds of negative and neutral self-images, respectively, of being reported, while the odds of reporting negative vs. neutral self-images were not statistically different from 1. We then performed a multinomial logistic regression to test whether age group, psychological well-being constructs and their interactions predicted the emotional valence of the first self-image. Life satisfaction was a significant predictor of the emotional valence of the first statement (*β* = 3.68, Standard error [*SE*] = 1.86, *z* = 1.98, *p* = 0.048, *r* = 0.22 [0.01, 0.44]): specifically, higher levels of satisfaction were associated with a higher likelihood of positive instead of negative first self-images. No other significant effects were found. See Appendix A for details.

In the next analysis, we used all the statements. This meant that we needed to take into account the nesting of observations in participants. Therefore, we specified a multilevel multinomial logistic regression, with the same predictors as before. Satisfaction with life was a significant predictor of the emotional valence of the self-images: specifically, higher levels of satisfaction with life were associated with a higher likelihood of neutral than negative self-images (*β* = 0.91, *SE* = 0.33, *z* = 2.74, *p* = 0.006, *r* = 0.30 [0.09, 0.51]) and a higher likelihood of positive than negative ones (*β* = 0.99, *SE* = 0.35, *z* = 2.78, *p* = 0.005, *r* = 0.30 [0.10, 0.51]). The likelihood of neutral rather than negative images was also higher for lower levels of positive affect (*β* = −1.09, *SE* = 0.40, *z* = −2.73, *p* = 0.006, *r* = −0.30 [−0.51, −0.09]). Interestingly, the main effects of Group and Group by Positive Affect interaction were also significant: specifically, younger adults were more likely compared to older adults to report neutral images rather than negative self-images (*β* = 0.87, *SE* = 0.44, *z* = 1.98, *p* = 0.048, *r* = 0.22 [0.01, 0.44]), and for higher levels of positive affect, younger adults were more likely compared to older adults to report neutral than negative self-images (*β* = 1.75, *SE* = 0.55, *z* = 3.20, *p* = 0.001, *r* = 0.34 [0.15, 0.54]) and positive than negative ones (*β* = 1.28, *SE* = 0.58, *z* = 2.21, *p* = 0.027, *r* = 0.25 [0.03, 0.46]). See Appendix A for details.

In the last analysis, we took into account the order of generation of the self-images. We created a variable whose categories identified the statements by their order (i.e., first, second, third, etc.) and dummy coded it setting the first statement as the reference. Therefore, the coefficients reported here and in Appendix A for the dummy variables of statements (i.e., st02, st03, st04, etc.) are the expected difference in the log odds of the emotional valence being in one category (e.g., neutral) compared to another (e.g., negative) for that statement with respect to the first one. For instance, the coefficient for st04 in the Positive vs. Neutral subtable is −1.25. This means that for the fourth statement the likelihood of being positive (instead of neutral) is smaller than for the first. In other words, the fifth statement had a higher likelihood of being neutral (instead of positive) with respect to the first one. We chose not to weight the statements according to their position in the rank order, with the first statement being assigned a highest value (i.e., 10) and with the last being assigned the lowest value (i.e., 1), because this coding would have meant that only a linear effect of position could have been tested, while with the dummy coding we could have also detected non-linear effects. 

The analyses carried out on the emotional valence of all statements considering statement as a fixed effect replicated almost completely the results about all statements. Specifically, life satisfaction was a significant predictor of the emotional valence of the self-images, with higher levels of satisfaction being associated with a higher likelihood of neutral than negative self-images (*β* = 0.94, *SE* = 0.36, *z* = 2.60, *p* = 0.009, *r* = 0.29 [0.08, 0.49]) and a higher likelihood of positive than negative ones (*β* = 1.04, *SE* = 0.40, *z* = 2.61, *p* = 0.009, *r* = 0.29 [0.08, 0.49]). The likelihood of neutral rather than negative images was also higher for lower levels of positive affect (*β* = −1.13, *SE* = 0.43, *z* = −2.61, *p* = 0.009, *r* = −0.29 [−0.49, −0.08]). The Group by Positive Affect interaction was also significant: for higher levels of positive affect, younger adults were more likely compared to older adults to report neutral than negative self-images (*β* = 1.78, *SE* = 0.59, *z* = 3.03, *p* = 0.002, *r* = 0.33 [0.13, 0.53]) and positive than negative ones (*β* = 1.29, *SE* = 0.64, *z* = 2.03, *p* = 0.042, *r* = 0.23 [0.01, 0.44])

We also observed a tendency of lower likelihood of reporting positive self-images instead of neutral ones from the fourth statement onwards: specifically, the regression coefficients of the fourth and fifth statements were negative and statistically significant (*β* = −1.25, *SE* = 0.41, *z* = −3.04, *p* = 0.002, *r* = −0.33 [−0.53, −0.13] and *β* = −1.54, *SE* = 0.42, *z* = −3.71, *p* < 0.001, *r* = −0.39 [−0.58, −0.20], respectively), with the coefficients of later statements, although no longer significant, still being negative (see Appendix A for details).

### 4.2. Personal Relevance

The same set of analyses was repeated for the personal relevance scores. In this case, given the answer scale, we considered relevance as an interval-level measure. The mean relevance score, when corrected for the nesting of observations in participants and statements, was 3.80 [3.67, 3.92]. Descriptive statistics for each statement in each group are reported in Table 3.

We specified a linear model to test whether age group, well-being constructs and their interactions predicted the personal relevance of the first self-image. Results showed that negative affect and life satisfaction significantly predicted the rating of personal relevance, with higher levels of negative affect and life satisfaction being associated with higher levels of personal relevance (*β* = 0.74, *SE* = 0.36, *z* = 2.07, *p* = 0.043, *r* = 0.25 [0.02, 0.48] and *β* = 0.57, *SE* = 0.22, *z* = 2.57, *p* = 0.013, *r* = 0.30 [0.08, 0.53], respectively). No other effects were significant. See Appendix A for details.

In order to test the association of age group, well-being measures and their interactions with the personal relevance of all statements, we specified a linear mixed model that took into account the nesting of observations in participants and statements. We did not find any significant effects (see Appendix A for details).

Finally, we specified a linear mixed model with statements as a fixed effect. We did not find any significant effects (see Appendix A for details).

### 4.3. Categories of Self-Images

We then analyzed the association of age group, well-being measures and their interactions with the category content of the statements. Globally, the most frequent themes were Traits (48.21%), Emotional states (12.62%), Specific attributes (8.10%), and Social identities (6.07%). The other five categories showed a relative frequency smaller than 5% (Evaluative descriptions: 4.64%; Nonsense: 3.93%; Peripheral information: 1.90%; Physical descriptions: 1.55%; Global descriptions: 0.95%) and were not considered in the analyses. Traits had 8.69 (*p* < 0.001), 5.52 (*p* < 0.001) and 3.93 (*p* < 0.001) times the odds of Social identities, Specific attributes, and Emotional states, respectively, of being reported, thereby suggesting that, in general, traits as self-images were more likely to be reported. The complete frequency table is reported in Table 4.

When we looked at the first self-image, given the low frequencies of Specific attributes and Social identities in the first statement, we could not carry out an analysis that took simultaneously into account age group and well-being measures as predictors of content category. However, a Fisher exact test on the crosstable Group by Content category considering only the four aforementioned most frequent categories was significant (*p* = 0.005). This association was due to young participants being more likely to report Traits compared to older adults. When we considered all the statements together and specified a multilevel multinomial regression model with age group, well-being measures and their interactions as predictors, the main effect of Group and the Positive Affect by Group interaction were significant predictors of content categories: specifically, younger adults were more likely than older adults to report traits as self-images instead of emotional states (*β* = 1.31, *SE* = 0.66, *z* = 1.99, *p* = 0.047, *r* = 0.23 [0.01, 0.44]). For higher level of positive affect, they were also less likely than older adults to report Emotional states and Traits instead of Specific attributes (*β* = −1.66, *SE* = 0.79, *z* = −2.10, *p* = 0.036, *r* = −0.24 [−0.45, −0.02] and *β* = −1.68, *SE* = 0.58, *z* = −2.92, *p* = 0.004, *r* = −0.32 [−0.53, −0.12], respectively). See Appendix A for details.

Finally, we specified a multilevel multinomial regression model with age group, well-being measures, their interactions, and statements as predictors. Consistent with the results of the analysis about all statements, the main effect of Group and the interaction Positive Affect by Group were significant: younger participants were more likely than older adults to report Traits instead of Social identities (*β* = 1.62, *SE* = 0.78, *z* = 2.09, *p* = 0.037, *r* = 0.24 [0.02, 0.45]) and instead of Emotional States (*β* = 1.48, *SE* = 0.46, *z* = 3.20, *p* = 0.001, *r* = 0.35 [0.15, 0.55]). For higher levels of positive affect, they were less likely than older adults to report Emotional states and Traits instead of Specific attributes (*β* = −1.63, *SE* = 0.82, *z* = −1.97, *p* = 0.049, *r* = −0.22 [−0.44, −0.01] and *β* = −1.66, *SE* = 0.68, *z* = −2.44, *p* = 0.015, *r* = −0.27 [−0.49, −0.06], respectively). Furthermore, later statements tended to be more likely that the first to be about Specific attributes or Traits instead of Social Identities and less likely than the first to be about Emotional states and Traits instead of Specific attributes. See Appendix A for details.

## 5. Discussion 

In the present study, we investigated whether and how age group (i.e., young vs. older adults), dimensions of psychological well-being (i.e., satisfaction with life, positive and negative affect and perceived stress), and their interactions predicted the phenomenological properties (i.e., emotional valence, personal relevance, category content) of semantic self-images. We also tested whether and how the different levels of accessibility of the images affected these patterns of associations. To this aim, we specified regression models for the first self-image, for all self-images regardless of their order of generation, and for all self-images taking into account their order of generation.

In general, the self-concepts generated were predominantly positive in both age groups and they were all rather personally relevant, as the lowest mean rating of relevance of self-statement was 3.36 out of 5. As for the content categories, traits as self-images were more likely to be reported compared to other categories.

The results on the emotional valence are consistent with previous studies carried out on samples of young and older adults [10,35], showing that semantic self-images were rated equally positively by adults in both age groups. 

The relatively high levels of personal relevance of self-statements are consistent with the instructions of the “Twenty-Statements” Test (TST), which requires participants to generate statements that describe themselves, namely aspects of their sense of identity, and they are in line with the results of previous studies in which the personal relevance of self-images was assessed [16]. 

Most important, we found that age group, psychological well-being, and their interactions predicted the phenomenological properties of semantic self-images and that the different levels of accessibility of self-images, that is, the order of generation of self-statements, also had some predictive power over and above the effects of the other predictors.

Of particular interest is the comparison between the patterns observed for the first self-image and for all self-images. When we considered the first self-image, independently of age group, satisfaction with life predicted the emotional valence of the statement, with higher levels of satisfaction associated with a higher likelihood of positive instead of negative self-images. Moreover, higher levels of satisfaction with life and negative affect were associated with higher ratings of personal relevance of statements. An age group effect was instead found in the category content of the first self-image, as young participants were more likely than elder participants to report Traits in their first self-statement.

When we took into account all statements, we found significant effects of age group and interaction of age group by measures of well-being on the emotional valence of self-images. Specifically, younger adults were more likely compared to older adults to report neutral images rather than negative self-images and, for higher levels of positive affect, they were more likely compared to older adults to report neutral than negative self-images and positive than negative ones. Moreover, consistent with the results obtained with the first self-image, higher levels of satisfaction with life were associated with higher likelihood of reporting positive instead or negative self-images. Effects of age group and interaction of age group by measures of well-being were also found on the content categories of self-images: young participants were more likely to report Traits instead of Emotional states than older participants, and for higher levels of positive affect, younger participants were less likely that older ones to report Emotional states and Traits instead of Specific attributes. No significant effect of the predictors was found for personal relevance of all self-images.

Finally, when we entered in the prediction model the statements as a fixed effect to test the effect of the order of generation, we observed a tendency of a lower likelihood of reporting positive self-images instead of neutral ones in later statements, which also differed from the first self-images in terms of content categories, as they tended to be more likely than the first one to refer to Specific attributes or Traits instead of Social Identities and less likely than the first to be about Emotional states and Traits instead of Specific attributes.

These findings, although preliminary, contribute to a growing body of research on semantic self-images and they improve our understanding of this topic in three main directions, namely (i) the association between semantic self-images and wellbeing, (ii) the effect of age group and the interaction effect of age group and well-being on semantic self-images, and (iii)the contribution of a dynamic, “accessibility-based” approach in the investigation of semantic self-images and their patterns of associations with background and psychological variables.

As for wellbeing, our findings replicate a strong association between well-being and semantic self-images, showing that dimensions of well-being, such as life satisfaction and positive and negative affect, predicted phenomenological properties of self-concepts. 

The relationship between wellbeing and the emotional valence of semantic self-image has been systematically examined in the study by Rathbone et al. [13]. In that seminal paper, the authors could show, for the first time, that dimensions of well-being were more closely linked to semantic self-image valence than episodic autobiographical memory valence and this effect was particularly pronounced in older adults. 

However, in that study, emotional valence was considered as a metric variable in which higher scores corresponded to higher levels of positive valence despite the answer scale being bipolar. Moreover, effects of interaction between age group and measures of well-being were not directly tested. In the present study, we took into account that the valence ratings might not represent a linear continuum by recoding scores into negative, neutral, and positive categories, and directly tested the interaction effect of age group and well-being dimensions on emotional valence, personal relevance, and category of the self-images.

As for the effect of age group on emotional valence, the main effect of age group was significant only in terms of younger participants being more likely than older ones to report neutral rather negative self-images, which does not suggest a “positivity effect” in younger adults. Indeed, semantic self-images were predominantly positive across both age groups, and this result is in line with previous studies, showing the same tendency in both younger and older adults to generate predominantly positive self-images (e.g., [10,35]). This pattern of results is consistent with the theory that most people have a tendency to be positive about the current self [34,57,58] and it suggests that the “positivity effect” in ageing reported for episodic autobiographical memories (e.g., [59]) may not extend to semantic self-images (see for a discussion, [13]). 

Interestingly, we also found a significant age group by positive affect interaction that indicated that, for higher levels of positive affect, younger participants were more likely than older ones to report neutral and positive self-images rather than negative ones, and less likely to report emotional states and traits rather than specific attributes. Although these results cannot conclusively establish whether age moderates the effect of well-being dimensions or vice versa, they nonetheless more convincingly suggest there are likely to be differences between young and older participants in the way that dimensions of well-being predict the emotional valence and content categories of sematic self-images, which deserve to be further investigated.

Another important contribution of our results comes from the application of a dynamic, “accessibility-based” approach in the assessment of semantic self-images. 

According to the Self-Memory-System (SMS), when participants are asked to generate self-statements in tasks like the “Twenty Statements” Test, they access a set of working self-conceptual knowledge. Some studies have demonstrated the flexibility of the working self-concepts, showing that different types of self-knowledge can be activated in different contexts, that is, the information we can access about ourselves and therefore the way we think about and describe ourselves can change depending on several factors [36,37,38,39].

Moreover, although all the statements generated at the task are accessible, they differ in terms of accessibility and the order of generation is supposed to reflect these differences. The issue of the accessibility seems to be particularly important when we compare the first image to the other ones. The particular status of the first self-image/self-statement comes from its intrinsic properties, being the earliest and most easily generated self-image, that is, the one with the highest level of accessibility and the only one which is not conditioned from other responses. Although the issue of the accessibility of self-images has been addressed only in a very few studies, and using different methods, when the order of generation has been considered it provided important information about the underlying process of generating self-images and about the differences among the self-images [14,16]. 

Our findings provided further support for the importance of an “accessibility-based” approach, as they showed a partially different pattern of effects of age and well-being when the analyses were carried out on the first image vs. all self-images, and also some apparent trend in the differences between the first statement and the following ones in predicting emotional valence and category content.

In evaluating these results, some limitations due to the preliminary nature of our data, as well as multiple avenues for future work, can be identified. Despite each participant providing ten data points, so that the analyses were performed on a dataset with at least 760 observations and some small (|*r*| < 0.30) effect sizes were statistically significant, we refrained from specifying other effects that could have been of interest, such as the interaction between statements and the other predictors, which would have allowed us to test whether and how the order of generation of statements moderated the effect of the other predictors (or vice versa). On a related note, the test of the differences between the first statement and the following ones clearly needed more statistical power to provide more convincing evidence, given the quite small (*r* in in the 0.10s) effect size.

However, since the data were collected during the pandemic, we chose to limit the duration of data collection to a period of moderate COVID-19 restrictions in our region and we matched our age samples in terms of several background variables, including COVID-19 restrictions, and levels of familiarity/knowledge with psychology topics. As for the group of older adults, in the present study we did not screen for cognitive impairment. Although it is unlikely that a person with a substantial cognitive impairment could successfully complete our protocol, in future replications a brief cognitive screening test might be included. 

Future studies are needed to replicate and extend our results in a pandemic transition phase (out of COVID-19 outbreaks) and to different and more representative samples of the population of interest. 

In the present study, the assessment of the semantic self-images and well-being was carried out online, remotely. Web-based methodology has several advantages [60] as it provides an efficient and cost-effective solution, especially when one is interested in comparing multiple measures taken on different days, as in the present study; during the pandemic, the possibility to run from remote increased the availability of testing and participants. 

Web-based assessment of cognitive functioning, including phenomenological aspects of autobiographical memories, is feasible also with older adults and in unsupervised settings [60,61,62] and in some studies older adults evaluated computer-based and web-based self-administered measures more favourably than paper-and-pencil tests, as they perceived these modalities as more user-friendly and less stressful [63,64,65]. Moreover, the flexibility of scheduling might contribute to a less taxing experience, increasing compliance and reducing the risk of dropout.

However, assessing remotely may also have some limitations. For instance, in our study we could not reliably measure the time needed to generate each statement, as we could not know for sure whether longer response times were due to the need of more time to answer, to a connection issue, or to the participant being distracted by some external stimulus.

Another important and theoretical limitation should be considered when interpreting these results. Given the cross-sectional nature of the current study, the regression coefficients of well-being can be interpreted only in terms of their predictive, and not causal, role. Longitudinal and experimental designs would be needed to examine the causal mechanisms between semantic self-image phenomenological properties and well-being. 

Finally, in the current study, we applied an “accessibility-based” approach in the investigation of current self-concepts in normal adult populations. Future studies will also benefit from an extension of this approach to the investigation of alterations of the self and their association with measures of well-being in clinical and neuropsychological samples, to have a finer-grained investigation of the changes in the self-concepts related to brain damage or psychopathology.

## Figures and Tables

**Table 1 brainsci-12-00716-t001:** Descriptive statistics of background variables and results of the comparisons between age groups.

Variable	Older Adults (*n* = 37)	Young (*n* = 47)	Statistic
Age (years, M ± SD)	66.54 ± 3.90	22.85 ± 2.40	*t*(56.52) = 59.90, *p* < 0.001, *d* = 13.52 [10.93, 15.99]
Females (proportion)	0.78	0.91	*χ*^2^(1) *=* 1.73, *p* = 0.189, *r* = 0.19 [0.00, 0.41]
Years of education (M ± SD)	15.50 ± 3.55	14.28 ± 1.57	*t*(39.55) = 1.82, *p* = 0.076, *d* = 0.44 [−0.05, 0.93]
Fear of COVID-19 Scale score (M ± SD, Ω)	16.13 ± 4.79, 0.75	16.87 ± 5.36, 0.84	*t*(66.94) = −0.62, *p* = 0.534, *d* = −0.14 [−0.60, 0.31]
SWLS score (M ± SD, Ω)	23.70 ± 6.82, 0.91	23.35 ± 6.72, 0.90	*t*(61.45) = 0.22, *p* = 0.826, *d* = 0.05 [−0.41, 0.51]
PANAS-P score (M ± SD, Ω)	26.37 ± 6.25, 0.87	28.63 ± 8.87, 0.91	*t*(73.49) = −1.30, *p* = 0.196, *d* = −0.30 [−0.74, 0.15]
PANAS-N score (M ± SD, Ω)	19.17 ± 6.71, 0.87	27.26 ± 8.27, 0.90	*t*(70.40) = −4.68, *p* < 0.001, *d* = −1.07 [−1.55, −0.59]
PSS score (M ± SD, Ω)	23.73 ± 5.56, 0.84	29.24 ± 6.51, 0.90	*t*(68.67) = −3.94, *p* < 0.001, *d* = −0.91 [−1.38, −0.43]
Experience of health issues in the last 3 months (proportion)	0.20	0.17	*χ*^2^(1) *<* 0.01, *p* > 0.999, *r* = 0.03 [0.00, 0.24]
Use of psychotropic drugs (proportion)	0.07	0.07	*χ*^2^(1) *<* 0.01, *p* > 0.999, *r* = 0.00 [0.00, 0.00]
Past anxiety disorders (proportion)	0.07	0.15	*χ*^2^(1) *=* 0.58, *p* = 0.444, *r* = 0.13 [0.00, 0.35]
Past depressive disorders (proportion)	0.20	0.11	*χ*^2^(1) *=* 0.60, *p* = 0.440, *r* = 0.13 [0.00, 0.35]

*Note*: M = mean; SD = standard deviation; Ω = omega reliability coefficient.

**Table 2 brainsci-12-00716-t002:** Crosstabulation of age group by emotional valence for each statement of the TST.

Statement 1		Valence		
Age group	Negative	Neutral	Positive	Total
Elderly	9 (24.3%)	10 (27.0%)	18 (48.6%)	37 (100.0%)
Young	10 (21.3%)	16 (34.0%)	21 (44.7%)	47 (100.0%)
Total	19 (22.6%)	26 (31.0%)	39 (46.4%)	84 (100.0%)
Statement 2				
Elderly	9 (25.0%)	10 (27.8%)	17 (47.2%)	36 (100.0%)
Young	9 (19.1%)	19 (40.4%)	19 (40.4%)	47 (100.0%)
Total	18 (21.7%)	29 (34.9%)	36 (43.4%)	83 (100.0%)
Statement 3				
Elderly	9 (24.3%)	7 (18.9%)	21 (56.8%)	37 (100.0%)
Young	8 (17.0%)	15 (31.9%)	24 (51.1%)	47 (100.0%)
Total	17 (20.2%)	22 (26.2%)	45 (53.6%)	84 (100.0%)
Statement 4				
Elderly	4 (11.1%)	15 (41.7%)	17 (47.2%)	36 (100.0%)
Young	6 (13.0%)	17 (37.0%)	23 (50.0%)	46 (100.0%)
Total	10 (12.2%)	32 (39.0%)	40 (48.8%)	82 (100.0%)
Statement 5				
Elderly	5 (13.9%)	18 (50.0%)	13 (36.1%)	36 (100.0%)
Young	9 (20.5%)	18 (40.9%)	17 (38.6%)	44 (100.0%)
Total	14 (17.5%)	36 (45.0%)	30 (37.5%)	80 (100.0%)
Statement 6				
Elderly	10 (28.6%)	9 (25.7%)	16 (45.7%)	35 (100.0%)
Young	11 (26.2%)	16 (38.1%)	15 (35.7%)	42 (100.0%)
Total	21 (27.3%)	25 (32.5%)	31 (40.3%)	77 (100.0%)
Statement 7				
Elderly	8 (22.9%)	13 (37.1%)	14 (40.0%)	35 (100.0%)
Young	6 (16.7%)	19 (52.8%)	11 (30.6%)	36 (100.0%)
Total	14 (19.7%)	32 (45.1%)	25 (35.2%)	71 (100.0%)
Statement 8				
Elderly	3 (8.6%)	13 (37.1%)	19 (54.3%)	35 (100.0%)
Young	7 (19.4%)	8 (22.2%)	21 (58.3%)	36 (100.0%)
Total	10 (14.1%)	21 (29.6%)	40 (56.3%)	71 (100.0%)
Statement 9				
Elderly	8 (24.2%)	13 (39.4%)	12 (36.4%)	33 (100.0%)
Young	5 (15.6%)	15 (46.9%)	12 (37.5%)	32 (100.0%)
Total	13 (20.0%)	28 (43.1%)	24 (36.9%)	65 (100.0%)
Statement 10				
Elderly	8 (24.2%)	13 (39.4%)	12 (36.4%)	33 (100.0%)
Young	5 (17.2%)	14 (48.3%)	10 (34.5%)	29 (100.0%)
Total	13 (21.0%)	27 (43.5%)	22 (35.5%)	62 (100.0%)

**Table 3 brainsci-12-00716-t003:** Descriptive statistics of the personal relevance score in each statement by age group.

	Age Group					
	Elderly			Young		
Statement	*n*	M	SD	*n*	M	SD
st01	36	3.78	1.07	47	3.85	0.96
st02	36	3.81	0.92	47	3.83	0.76
st03	37	3.68	1.00	47	3.85	0.83
st04	36	3.72	0.91	46	3.89	0.77
st05	36	3.36	0.99	44	3.86	0.82
st06	35	3.54	0.89	42	3.71	0.94
st07	35	3.49	1.12	36	4.06	0.89
st08	35	3.74	0.85	36	4.00	0.76
st09	33	3.73	0.94	32	3.69	0.93
st10	33	3.67	0.96	29	3.79	0.82

Note: *n* = number of valid scores; M = mean; SD = standard deviation.

**Table 4 brainsci-12-00716-t004:** Crosstabulation of age group by content category for each statement of the TST.

	Content Category									
Statement 1	Traits	Emotional States	Specific Attributes	Social Identities	Evaluative Descriptions	Nonsense	Peripheral Information	Physical Descriptions	Global Descriptions	Total
Elderly	10 (28.6%)	9 (25.7%)	1 (2.9%)	7 (20.0%)	2 (5.7%)	3 (8.6%)	1 (2.9%)	0 (0.0%)	2 (5.7%)	35 (100.0%)
Young	30 (63.8%)	7 (14.9%)	1 (2.1%)	2 (4.3%)	3 (6.4%)	1 (2.1%)	1 (2.1%)	1 (2.1%)	1 (2.1%)	47 (100.0%)
Total	40 (48.8%)	16 (19.5%)	2 (2.4%)	9 (11.0%)	5 (6.1%)	4 (4.9%)	2 (2.4%)	1 (1.2%)	3 (3.7%)	82 (100.0%)
Statement 2										
Elderly	12 (35.3%)	9 (26.5%)	2 (5.9%)	4 (11.8%)	1 (2.9%)	4 (11.8%)	0 (0.0%)	2 (5.9%)	0 (0.0%)	34 (100.0%)
Young	30 (63.8%)	9 (19.1%)	3 (6.4%)	3 (6.4%)	2 (4.3%)	0 (0.0%)	0 (0.0%)	0 (0.0%)	0 (0.0%)	47 (100.0%)
Total	42 (51.9%)	18 (22.2%)	5 (6.2%)	7 (8.6%)	3 (3.7%)	4 (4.9%)	0 (0.0%)	2 (2.5%)	0 (0.0%)	81 (100.0%)
Statement 3										
Elderly	14 (40.0%)	6 (17.1%)	4 (11.4%)	4 (11.4%)	4 (11.4%)	3 (8.6%)	0 (0.0%)	0 (0.0%)	0 (0.0%)	35 (100.0%)
Young	31 (66.0%)	4 (8.5%)	5 (10.6%)	5 (10.6%)	2 (4.3%)	0 (0.0%)	0 (0.0%)	0 (0.0%)	0 (0.0%)	47 (100.0%)
Total	45 (54.9%)	10 (12.2%)	9 (11.0%)	9 (11.0%)	6 (7.3%)	3 (3.7%)	0 (0.0%)	0 (0.0%)	0 (0.0%)	82 (100.0%)
Statement 4										
Elderly	14 (41.2%)	8 (23.5%)	1 (2.9%)	5 (14.7%)	2 (5.9%)	4 (11.8%)	0 (0.0%)	0 (0.0%)	0 (0.0%)	34 (100.0%)
Young	28 (60.9%)	6 (13.0%)	3 (6.5%)	4 (8.7%)	0 (0.0%)	1 (2.2%)	4 (8.7%)	0 (0.0%)	0 (0.0%)	46 (100.0%)
Total	42 (52.5%)	14 (17.5%)	4 (5.0%)	9 (11.2%)	2 (2.5%)	5 (6.2%)	4 (5.0%)	0 (0.0%)	0 (0.0%)	80 (100.0%)
Statement 5										
Elderly	17 (50.0%)	4 (11.8%)	4 (11.8%)	3 (8.8%)	4 (11.8%)	1 (2.9%)	0 (0.0%)	1 (2.9%)	0 (0.0%)	34 (100.0%)
Young	24 (54.5%)	9 (20.5%)	4 (9.1%)	2 (4.5%)	2 (4.5%)	0 (0.0%)	2 (4.5%)	0 (0.0%)	1 (2.3%)	44 (100.0%)
Total	41 (52.6%)	13 (16.7%)	8 (10.3%)	5 (6.4%)	6 (7.7%)	1 (1.3%)	2 (2.6%)	1 (1.3%)	1 (1.3%)	78 (100.0%)
Statement 6										
Elderly	18 (54.5%)	3 (9.1%)	3 (9.1%)	3 (9.1%)	1 (3.0%)	2 (6.1%)	2 (6.1%)	0 (0.0%)	1 (3.0%)	33 (100.0%)
Young	30 (71.4%)	2 (4.8%)	5 (11.9%)	2 (4.8%)	0 (0.0%)	1 (2.4%)	1 (2.4%)	1 (2.4%)	0 (0.0%)	42 (100.0%)
Total	48 (64.0%)	5 (6.7%)	8 (10.7%)	5 (6.7%)	1 (1.3%)	3 (4.0%)	3 (4.0%)	1 (1.3%)	1 (1.3%)	75 (100.0%)
Statement 7										
Elderly	12 (36.4%)	8 (24.2%)	3 (9.1%)	2 (6.1%)	3 (9.1%)	2 (6.1%)	1 (3.0%)	2 (6.1%)	0 (0.0%)	33 (100.0%)
Young	23 (63.9%)	3 (8.3%)	4 (11.1%)	0 (0.0%)	2 (5.6%)	2 (5.6%)	0 (0.0%)	1 (2.8%)	1 (2.8%)	36 (100.0%)
Total	35 (50.7%)	11 (15.9%)	7 (10.1%)	2 (2.9%)	5 (7.2%)	4 (5.8%)	1 (1.4%)	3 (4.3%)	1 (1.4%)	69 (100.0%)
Statement 8										
Elderly	16 (48.5%)	5 (15.2%)	5 (15.2%)	2 (6.1%)	0 (0.0%)	2 (6.1%)	1 (3.0%)	1 (3.0%)	1 (3.0%)	33 (100.0%)
Young	20 (55.6%)	3 (8.3%)	7 (19.4%)	0 (0.0%)	2 (5.6%)	1 (2.8%)	1 (2.8%)	2 (5.6%)	0 (0.0%)	36 (100.0%)
Total	36 (52.2%)	8 (11.6%)	12 (17.4%)	2 (2.9%)	2 (2.9%)	3 (4.3%)	2 (2.9%)	3 (4.3%)	1 (1.4%)	69 (100.0%)
Statement 9										
Elderly	15 (48.4%)	4 (12.9%)	7 (22.6%)	2 (6.5%)	1 (3.2%)	1 (3.2%)	1 (3.2%)	0 (0.0%)	0 (0.0%)	31 (100.0%)
Young	21 (65.6%)	5 (15.6%)	3 (9.4%)	0 (0.0%)	1 (3.1%)	0 (0.0%)	0 (0.0%)	1 (3.1%)	1 (3.1%)	32 (100.0%)
Total	36 (57.1%)	9 (14.3%)	10 (15.9%)	2 (3.2%)	2 (3.2%)	1 (1.6%)	1 (1.6%)	1 (1.6%)	1 (1.6%)	63 (100.0%)
Statement 10										
Elderly	19 (61.3%)	2 (6.5%)	1 (3.2%)	1 (3.2%)	4 (12.9%)	3 (9.7%)	0 (0.0%)	1 (3.2%)	0 (0.0%)	31 (100.0%)
Young	21 (72.4%)	0 (0.0%)	2 (6.9%)	0 (0.0%)	3 (10.3%)	2 (6.9%)	1 (3.4%)	0 (0.0%)	0 (0.0%)	29 (100.0%)
Total	40 (66.7%)	2 (3.3%)	3 (5.0%)	1 (1.7%)	7 (11.7%)	5 (8.3%)	1 (1.7%)	1 (1.7%)	0 (0.0%)	60 (100.0%)

## Data Availability

The data presented in this study are available on request from the corresponding authors. The data are not publicly available as stated in the informed consent form that participants signed.

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
