# Peer review of "Semantic Self-Images and Well-Being in Young and Older Adults: Does the Accessibility Matter?"

_brainsci, 2022, doi:10.3390/brainsci12060716_

Round 1

Reviewer 1 Report

This ms investigates how aging, dimensions of well-being and their interactions predicts phenomenological properties of semantic self-images. Results demonstrate how life satisfaction predicts positive and personal-relevant self-images.

The study is well designed, analyses are well conducted and the findings are of relevant, I only have one major comment and some minor comments to make, illustrated below:

1- Major comment: while exclusion criteria included general health issues & anxiety/depression, the participants, especially the older group, were not screened against cognitive decline. This screening should have included, at the very least, a brief general cognitive functioning test (e.g., MMSE/MOCA). You may wish to acknowledge this shortcoming in the discussion in case of you have no data about the participants’ cognitive functioning.

2- Minor comment: the Covid fear comes out of the bleu in the method section. You may wish to introduce this variable in the introduction section.

3- Minor comment: section 3.4.: can you provide some examples about the coding of statement? It is important to illustrate how a given statement was coded as trait, social identity, specific attribute…

By the way, although the introduction relates these self-images categories to previous literature, the authors are invited to better justify the use of these self-images categories (traits, social identities, specific attributes, evaluative descriptions, physical descriptions, emotional states, peripheral information, and global descriptions). Although the used method offers a rich capture of self-images, simplest, but robust, classification would be used such as the simple distinction between three basic self-dimensions (i.e., physical self, social self, vs psychological self). The latter distinction has been used in research in aging:

El Haj, M., Gallouj, K., & Antoine, P. (2019). Autobiographical recall as a tool to enhance the sense of self in Alzheimer’s disease. Archives of Gerontology and Geriatrics, 82, 28-34. https://doi.org/https://doi.org/10.1016/j.archger.2019.01.011

4- Minor comment: the finding that self-concepts generated were predominantly positive can be discussed in light of the well-known Carstensen’s work about the positive emotional shift in normal aging.

Reviewer 2 Report

Thank you for the opportunity to review this paper, describing a comparison of the self-descriptive statements of younger and older adults, and their relationship to wellbeing as well as accessibility. They report a large number of analyses which indicate some age differences in content and in emotional quality, as well as effects of well-being.

Overall, I believe the paper is interesting and worthwhile, since semantic aspects of autobiography have been studied little compared to episodic aspects. 

However, I found the analysis and the results extremely challenging to follow and to understand. There are a very large number of analyses presented, and it is challenging to draw general findings from them. There are many significant tests and no correction for multiple testing, which makes it hard to know how to evaluate the findings.  Is it possible to streamline the results to focus more squarely on key questions and comparisons, and reduce the number of statistical tests being conducted?

I was concerned about the sample size and power issues, especially as it appears that 7 of the 37 older adults were not included in the analysis due to missing data. Although this is acknowledged in the discussion, it provides an additional reason to reduce the number of tests and focus more on specific analyses where there are hypotheses about the direction of the outcome. 

I'm not sure what the results mean. Overall it appears that younger adults had more neutral than negative self-descriptors compared to older adults (although I note this is based on p = .048, so it is not very compelling as the major age difference in the study), and were more positive when they also reported higher levels of positive affect. How does this align with the finding in this study (and widely in the literature) of a positivity bias among older adults?

In summary, I think this study is interesting and has potential. But based on the current analyses and they way they are presented, I'm not sure what these data, in their current form, help us to understand or explain. 

I hope these comments are helpful.
